# Nucleosome-CHD4 chromatin remodeler structure maps human disease mutations

**Lucas Farnung\*, Moritz Ochmann, Patrick Cramer\***

Max Planck Institute for Biophysical Chemistry, Department of Molecular Biology, Göttingen, Germany

**Abstract** Chromatin remodeling plays important roles in gene regulation during development, differentiation and in disease. The chromatin remodeling enzyme CHD4 is a component of the NuRD and ChAHP complexes that are involved in gene repression. Here, we report the cryo-electron microscopy (cryo-EM) structure of *Homo sapiens* CHD4 engaged with a nucleosome core particle in the presence of the non-hydrolysable ATP analogue AMP-PNP at an overall resolution of 3.1 Å. The ATPase motor of CHD4 binds and distorts nucleosomal DNA at superhelical location (SHL) +2, supporting the 'twist defect' model of chromatin remodeling. CHD4 does not induce unwrapping of terminal DNA, in contrast to its homologue Chd1, which functions in gene activation. Our structure also maps CHD4 mutations that are associated with human cancer or the intellectual disability disorder Sifrim-Hitz-Weiss syndrome.

## Introduction

In the nucleus of eukaryotic cells, DNA is compacted into chromatin. The fundamental building block of chromatin is the nucleosome, a complex of ~146 base pairs (bp) of DNA wrapped around an octamer of histone proteins. The degree of chromatin compaction influences DNA replication, transcription, and repair. Maintenance of the appropriate chromatin state requires ATP-dependent chromatin-remodeling enzymes. These 'chromatin remodelers' are divided into four families called CHD, SWI/SNF, ISWI, and INO80 (*Clapier et al., 2017*). All chromatin remodelers contain a conserved ATPase core that hydrolyses ATP to alter contacts between nucleosomal DNA and the histone octamer and to facilitate nucleosome assembly, sliding, ejection, or histone exchange.

Members of the CHD ('chromodomain helicase DNA-binding') family of chromatin remodelers all contain a central SNF2-like ATPase motor domain and a double chromodomain in their N-terminal region. The double chromodomain binds modified histones (*Sims et al., 2005*) and interacts with nucleosomal DNA to regulate ATPase activity (*Nodelman et al., 2017*). Recent structures of the yeast remodeler Chd1 in complex with a nucleosome uncovered the architecture of one subfamily of CHD remodelers (subfamily I) and its interactions with the nucleosome (*Farnung et al., 2017*; *Sundaramoorthy et al., 2018*). A unique feature of these structures is that Chd1 binding induces unwrapping of terminal DNA from the histone octamer surface at superhelical location (SHL) −6 and −7 (*Farnung et al., 2017*; *Sundaramoorthy et al., 2018*). However, the resolution of these studies was limited, such that atomic details were not resolved.

The human CHD family member CHD4 (*Woodage et al., 1997*) shows nucleosome spacing activity (*Silva et al., 2016*). CHD4 is also known as Mi-2 in *Drosophila melanogaster* (*Kehle et al., 1998*). CHD4, CHD3, and CHD5 form CHD subfamily II, which differs in domain architecture from subfamily I. CHD3, CHD4, and CHD5 contain two N-terminal plant homeodomain (PHD) zinc finger domains (*Schindler et al., 1993*), a DNA-interacting double chromodomain, and the ATPase motor. CHD4 contains an additional high mobility group (HMG) box-like domain in its N-terminal region (*Silva et al., 2016*) and two additional domains of unknown function that are located in the C-terminal region.

**\*For correspondence:**
Lucas.Farnung@mpibpc.mpg.de (LF);
patrick.cramer@mpibpc.mpg.de (PC)

**Competing interests:** The authors declare that no competing interests exist.

CHD4 is implicated in the repression of lineage-specific genes during differentiation (*Liang et al., 2017*) and is required for the establishment and maintenance of more compacted chromatin structures (*Bornelöv et al., 2018*). CHD4 mutations have a high incidence in some carcinomas (*Kandoth et al., 2013*) and in thyroid and ovarian cancers (*Längst and Manelyte, 2015*). Mutations in CHD4 have also been implicated in intellectual disability syndromes (*Sifrim et al., 2016*; *Weiss et al., 2016*).

CHD4 is a subunit of the multi-subunit Nucleosome Remodeling Deacetylase (NuRD) complex (*Tong et al., 1998*; *Xue et al., 1998*; *Zhang et al., 1998*). NuRD also contains the deacetylase HDAC1/2 and accessory subunits that serve regulatory and scaffolding roles. NuRD is implicated in gene silencing, but also gene activation (*Gnanapragasam et al., 2011*). It is essential for cell cycle progression (*Polo et al., 2010*), DNA damage response (*Larsen et al., 2010*; *Smeenk et al., 2010*), establishment of heterochromatin (*Sims and Wade, 2011*), and differentiation (*Bornelöv et al., 2018*; *Burgold et al., 2019*). In addition, CHD4 is part of the heterotrimeric ChAHP complex that is also involved in gene repression (*Ostapcuk et al., 2018*).

Thus far, structural studies of CHD4 have been limited to individual domains (*Kwan et al., 2003*; *Mansfield et al., 2011*). Here, we report the cryo-electron microscopy (cryo-EM) structure of human CHD4 bound to a nucleosome at an overall resolution of 3.1 Å. CHD4 engages the nucleosome at SHL +2 and induces a conformational change in DNA at this location in the presence of the ATP analogue adenylyl imidodiphosphate (AMP-PNP). Structural comparisons show that CHD4, in contrast to Chd1, does not induce unwrapping of terminal DNA, and this is also observed in biochemical assays. Maintenance of the integrity of the nucleosome in the presence of CHD4 is consistent with the role of CHD4 in gene repression, and in heterochromatin formation and maintenance. Finally, the detailed nucleosome-CHD4 structure enables mapping of known human disease mutations (*Kovač et al., 2018*; *Sifrim et al., 2016*; *Weiss et al., 2016*) and indicates how these may perturb enzyme function.

## Results

### Nucleosome-CHD4 complex structure

To investigate how the human chromatin remodeller CHD4 engages a nucleosome, we determined the structure of *H. sapiens* CHD4 bound to a *Xenopus laevis* nucleosome core particle in the presence of the ATP analogue AMP-PNP. We recombinantly expressed and purified full-length CHD4 and reconstituted a complex of CHD4 with a pre-assembled nucleosome core particle. The nucleosome comprised 145 base pairs (bp) of DNA, corresponding to the Widom 601 sequence (*Lowary and Widom, 1998*) with additional 4 and 30 bp of extranucleosomal DNA on the entry and exit side of the nucleosome, respectively. The nucleosome-CHD4 complex was purified by size exclusion chromatography (*Figure 1—figure supplement 1*).

To determine the structure of the nucleosome-CHD4 complex, we collected single particle cryo-EM data on a Titan Krios (FEI) microscope equipped with a K2 direct electron detector (Gatan) (Materials and methods). We obtained a cryo-EM reconstruction of the nucleosome-CHD4 complex at an overall resolution of 3.1 Å (FSC 0.143 criterion) (*Figure 1—figure supplements 2–4*, *Video 1*). The nucleosome was resolved at a resolution of 3.0–4.5 Å, whereas CHD4 was resolved at 3.1–5.0 Å, depending on the protein region. The register of the DNA was unambiguously determined based on distinct densities for purine and pyrimidine nucleotides around the dyad (*Figure 1—figure supplement 3h*). Well-defined density was also obtained for AMP-PNP and a coordinated magnesium ion in the CHD4 active site (*Figure 1—figure supplement 3i*). The model was locally adjusted and

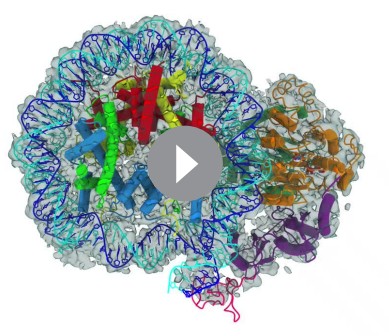

**Video 1.** Cryo-EM density and structure of the nucleosome-CHD4 complex.
https://elifesciences.org/articles/56178#video1

real-space refined, leading to very good stereochemistry (Materials and methods) (*Table 1*).

## CHD4 architecture

The CHD4 ATPase motor binds the nucleosome at SHL +2 (*Figure 1*, *Figure 1—figure supplement 4f*). Binding at this location has also been observed for the chromatin remodelers Chd1 (*Farnung et al., 2017*; *Sundaramoorthy et al., 2018*), Snf2 (*Liu et al., 2017*), and Swr1 (*Willhoft et al., 2018*). The ATPase motor is in a closed, post-translocated state with AMP-PNP bound in the active site. The same state and a similar conformation was observed for Chd1 when bound to ADP·BeF₃ (*Farnung et al., 2017*; *Sundaramoorthy et al., 2018*; *Sundaramoorthy et al., 2017*). The double chromodomain is located at SHL +1 and contacts the nucleosomal DNA phosphate backbone via electrostatic interactions, in a fashion similar to that observed for *S. cerevisiae* Chd1 (*Figure 1*; *Farnung et al., 2017*; *Nodelman et al., 2017*).

**Table 1.** Cryo-EM data collection, refinement and validation statistics.

| | Nucleosome-CHD4 complex (EMD-10058) (PDB 6RYR) | Nucleosome-CHD4₂ complex (EMDB-10059) (PDB 6RYU) |
|---|---|---|
| Data collection and processing | | |
| Magnification | 130,000 | 130,000 |
| Voltage (kV) | 300 | 300 |
| Electron exposure (e–/Å²) | 43–45 | 43–45 |
| Defocus range (μm) | 0.25–4 | 0.25–4 |
| Pixel size (Å) | 1.05 | 1.05 |
| Symmetry imposed | C1 | C1 |
| Initial particle images (no.) | 650,599 | 650,599 |
| Final particle images (no.) | 89,623 | 40,233 |
| Map resolution (Å) FSC threshold | 3.1 0.143 | 4.0 0.143 |
| Map resolution range (Å) | 3.0–5 | 3.7–8.3 |
| Refinement | | |
| Initial models used (PDB code) | 3LZ0, 5O9G, 2L75, 4O9I, 6Q3M | 3LZ0, 5O9G, 2L75, 4O9I, 6Q3M |
| Map sharpening *B* factor (Å²) | −36 | −86 |
| Model composition Non-hydrogen atoms Protein residues Nucleotides Ligands | 17,834 1463 298 4 | 23,598 2180 298 8 |
| *B* factors (Å²) Protein Nucleotide Ligand | 45.28 71.82 60.10 | 95.29 112.27 125.7 |
| R.m.s. deviations Bond lengths (Å) Bond angles (°) | 0.003 0.638 | 0.005 1.028 |
| Validation MolProbity score Clashscore Poor rotamers (%) | 1.54 5.69 0.08 | 1.92 6.52 1.64 |
| Ramachandran plot Favored (%) Allowed (%) Disallowed (%) | 96.50 3.50 0.0 | 94.16 5.84 0.0 |

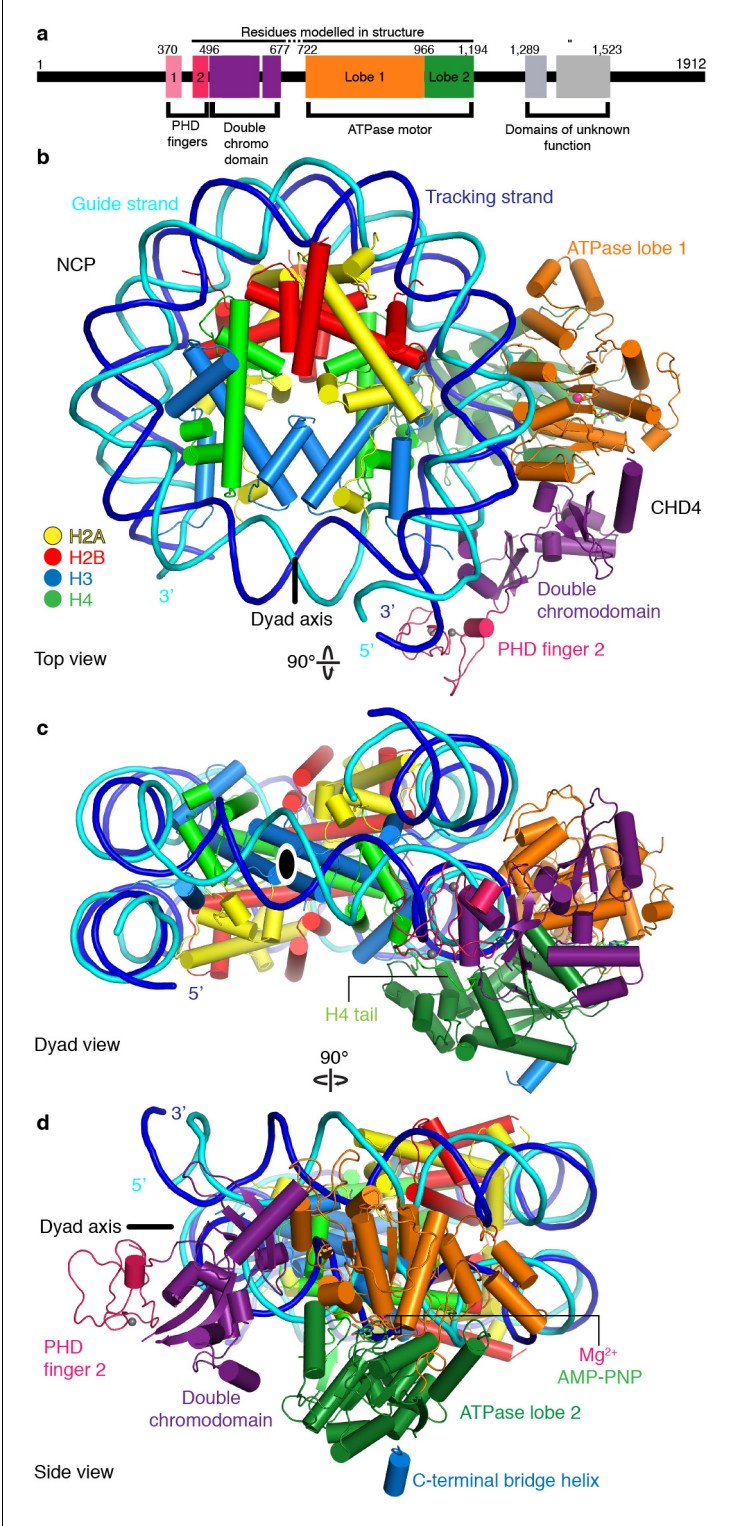

**Figure 1.** Structure of the nucleosome-CHD4 complex. (a) Schematic of domain architecture of CHD4. Domain borders are indicated. (b-d) Cartoon model viewed from the top (b), dyad (c), and side (d). Histones H2A, H2B, H3, H4, tracking strand, guide strand, CHD4 PHD finger 2, double chromodomain, ATPase lobe 1, and ATPase lobe 2 are colored in yellow, red, light blue, green, blue, cyan, pink, purple, orange, and forest green, respectively. Color code used throughout. The dyad axis is indicated as a black line or a black oval circle. Magnesium and zinc ions shown as pink and grey spheres, respectively. AMP-PNP shown in stick representation.

*Figure 1 continued on next page*

*Figure 1 continued*

The online version of this article includes the following figure supplement(s) for figure 1:

**Figure supplement 1.** Formation of the nucleosome-CHD4 complex.
**Figure supplement 2.** Cryo-EM structure determination.
**Figure supplement 3.** Cryo-EM densities.
**Figure supplement 4.** Data quality and metrics.

The PHD finger 2 of CHD4 is located near SHL +0.5 and the double chromodomain. This is consistent with NMR studies that predicted binding of this PHD finger close to the dyad axis and the H3 tail (*Gatchalian et al., 2017*). Additionally, we observe parts of the C-terminal bridge (*Hauk et al., 2010*), an amino acid segment that follows the ATPase lobes. Part of the C-terminal bridge docks against ATPase lobe 2 and extends toward the first ATPase lobe (*Figure 1*, *Figure 1—figure supplement 3j*). This region was not resolved in the nucleosome-Chd1 structures but was observed in a previously published crystal structure of auto-inhibited Chd1 (*Hauk et al., 2010*). Taken together, CHD4 and Chd1 share a core architecture that involves the ATPase motor and the double chromodomain but differ in their peripheral subfamily-specific protein features.

## CHD4 does not detach exit side nucleosomal DNA

In contrast to the nucleosome-Chd1 structure (*Farnung et al., 2017*), we did not observe unwrapping of nucleosomal DNA from the histone octamer on the second DNA gyre at SHL −6 and −7 (*Figure 2a*). To test whether this structural difference can be recapitulated biochemically in solution, we used a Förster Resonance Energy Transfer (FRET) assay to monitor putative DNA unwrapping activity by these two chromatin remodellers. The DNA 5′ ends of the nucleosome were labelled with Cy3 or Cy5 (*Figure 2b*). Using the doubly labeled nucleosome, FRET efficiencies were measured in the absence and presence of *S. cerevisiae* Chd1 (residues 1–1247) or full-length *H. sapiens* CHD4, and in the presence of AMP-PNP or ADP·BeF$_3$.

In these assays, Chd1 showed an increase in fluorescence emission of the donor and a reduction in the acceptor emission (*Figure 2c*). This indicated that the distance between the two DNA ends of the nucleosome increased upon Chd1 addition, and was consistent with the structurally observed DNA unwrapping of terminal DNA. In contrast, fluorescence emissions measured for the CHD4 sample did not differ from the nucleosome controls (*Figure 2c*), showing that CHD4 was unable to unwrap nucleosomal DNA both in the presence of AMP-PNP or ADP·BeF$_3$.

The major difference in DNA unwrapping between these two remodelers may be due to a lack of a DNA-binding region in CHD4, when compared to Chd1. Chd1 uses its DNA-binding region to interact extensively with terminal DNA on the exit side at SHL −7, and such contacts are absent in the nucleosome-CHD4 structure (*Figure 2*). It is likely that other CHD family members from subfamily II such as CHD3 and CHD5, which also lack a DNA-binding region, will also not induce unwrapping of terminal DNA.

## CHD4-DNA interactions

The high resolution of our nucleosome-CHD4 structure enables a detailed description of the interactions of the ATPase motor with nucleosomal DNA. CHD4 contacts the phosphate backbone of the tracking and guide strands via electrostatic interactions that are mediated by lysine and arginine residues (*Figure 3*). These interactions with the DNA phosphate backbone are formed by residues in the canonical ATPase motifs Ia, Ic, II, IV, IVa, V, and Va and by residues present in non-canonical motifs (e.g. Lys810) (*Figure 3*, *Figure 3—figure supplement 1*).

We also observe that residues Asn1010, Arg1127, and Trp1148 insert into the DNA minor groove over a stretch of seven base pairs (*Figure 3c*). Asn1010 is not part of a canonical ATPase motif and inserts into the DNA minor groove around SHL +2.5. Arg1127 (motif V) is universally conserved in all CHD chromatin remodelers and inserts into the DNA minor groove at SHL +2. Our density is consistent with two alternative conformations of the Arg1127 side chain, with the guanidinium head group pointing either toward the tracking or the guide strand of DNA. Trp1148 is located in motif Va, inserts into the minor groove near the guide strand, and plays a critical role in coupling ATPase hydrolysis and DNA translocation (*Liu et al., 2017*). We further observe a contact between a

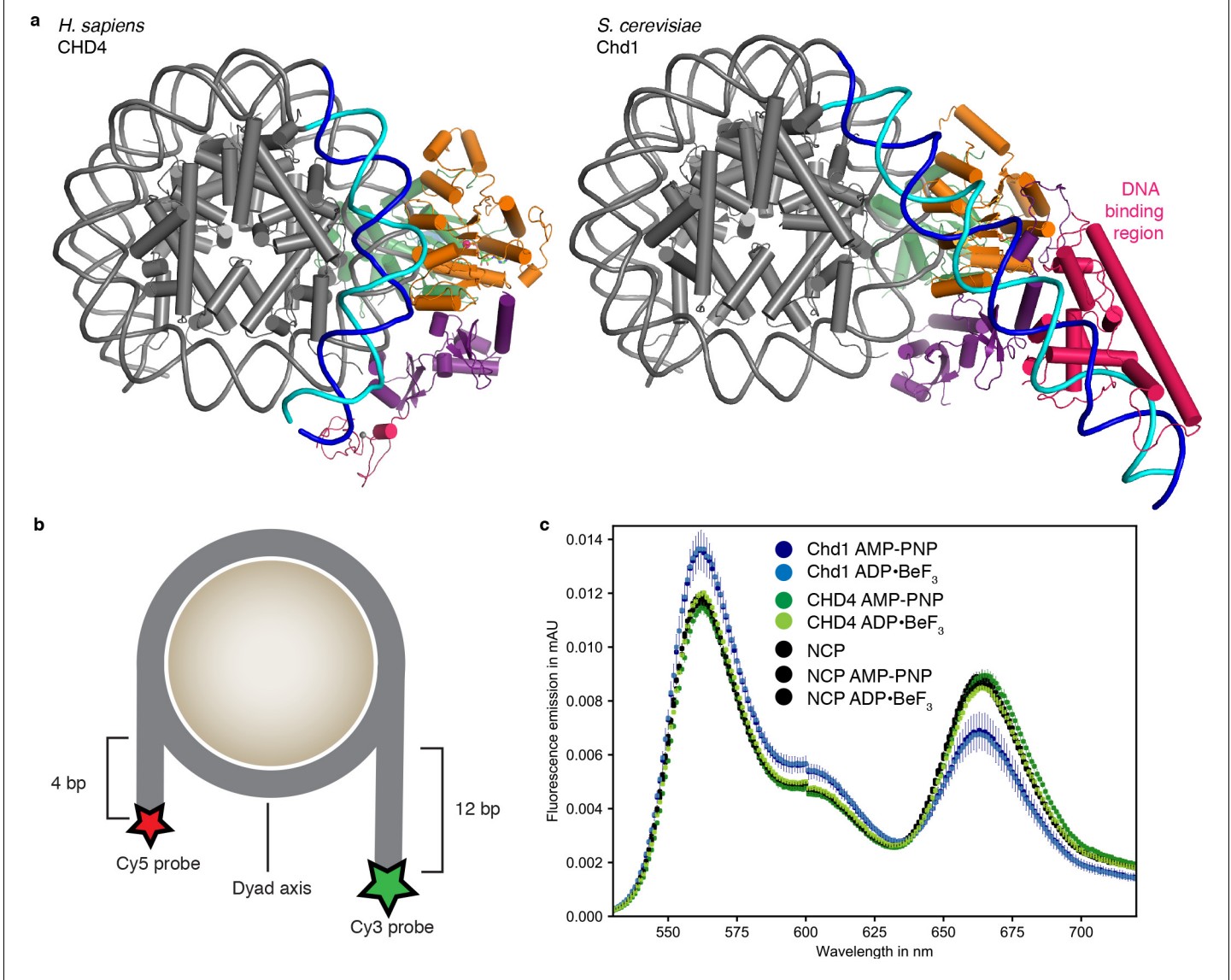

**Figure 2.** Comparison with nucleosome-Chd1 structure. (**a**) CHD4 (left) does not possess a DNA-binding region and does not detach DNA from the second gyre. Chd1 (right) detaches DNA from SHL −7 to −5, stabilizes the detached DNA via its DNA-binding region, and introduces a ~ 60° bend with respect to the canonical DNA position observed in the nucleosome-CHD4 structure. (**b**) Schematic of experimental FRET setup. (**c**) Fluorescence emission spectra produced after excitation at 510 nm of Cy3/Cy5 labeled nucleosome in the presence of *S. cerevisiae* Chd1 (residues 1–1247) or *H. sapiens* CHD4 and AMP-PNP or ADP·BeF$_3$ show unwrapping of nucleosomal DNA by Chd1 but not by CHD4.

The online version of this article includes the following source data for figure 2:

**Source data 1.** FRET source data.

positively charged loop in ATPase lobe 1 (residues 832–837) and the second DNA gyre at SHL −6. This loop is present in CHD3, CHD4, and CHD5, but not in Snf2 or ISWI remodelers (*Figure 3—figure supplement 1*).

## CHD4 binding distorts DNA at SHL +2

Comparison of our structure with a high-resolution X-ray structure of the free nucleosome (*Vasudevan et al., 2010*) reveals a conformational change in the DNA where the ATPase motor engages its DNA substrate (SHL +2) (*Figure 3d*). The high resolution of the nucleosome-CHD4 structure shows that ~5 DNA base pairs between SHL +1.5 and SHL +2.5 are pulled away from the octamer surface by up to 3 Å. This distortion does not include the previously observed 'bulging' or a

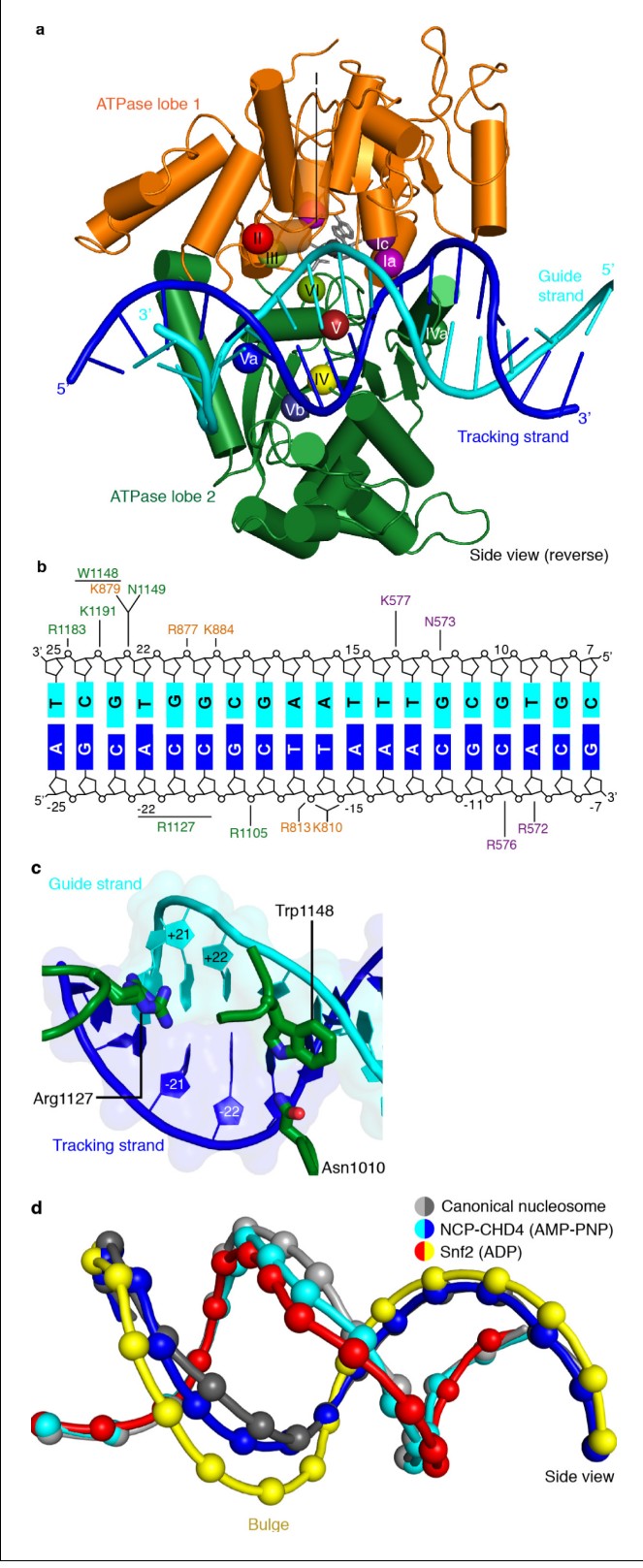

**Figure 3.** CHD4-DNA interactions and DNA distortion. (**a**) CHD4 interacts extensively with nucleosomal DNA around SHL +2. ATPase lobe 1 and lobe 2 of CHD4 are shown. Guide and tracking strands are indicated. ATPase motifs are shown as colored spheres and labelled. (**b**) Schematic depiction of DNA interactions of the double chromodomain, ATPase lobe 1 and lobe 2. (**c**) Asn1010, Trp1148 and Arg1227 insert into the minor groove
*Figure 3 continued on next page*

*Figure 3 continued*

between DNA tracking and guide strand. The two conformations of the Arg1127 side chain are shown. Nucleic acids are shown as cartoons with their respective surfaces. (d) Detailed cartoon representation of DNA distortion at SHL +2. Canonical nucleosome (PDB code 3LZ0, grey), AMP-PNP bound NCP-CHD4 structure (this study, blue and cyan), and ADP bound nucleosome-Snf2 structure (PDB code 5Z3O, red and yellow) are shown. Phosphate atoms shown as spheres.

The online version of this article includes the following figure supplement(s) for figure 3:

**Figure supplement 1.** Comparison of CHD4 with Chd1 and other chromatin remodelers.

___

'twist defect' that is characterized by a 1 bp local underwinding of the DNA duplex and observed when the ATPase motor adopts the open/apo or ADP-bound states (*Li et al., 2019*). In contrast, the DNA distortion observed in our AMP-PNP-bound state is an intermediate between the bulged and the canonical DNA conformation (*Figure 3d*). Such an AMP-PNP-bound intermediate DNA state was predicted based on biochemical experiments (*Winger et al., 2018*). This observation demonstrates that the extent of DNA distortion at SHL +2 depends on the functional state of the ATPase motor and is consistent with the proposed twist defect propagation model of chromatin remodeling (*Winger et al., 2018*).

## CHD4 binds the histone H4 tail

As observed for *S. cerevisiae* Chd1 (*Farnung et al., 2017*), *H. sapiens* CHD4 contacts the histone H4 tail with its ATPase lobe 2. The H4 tail is located between ATPase lobe 2 and the nucleosomal DNA at SHL +1.5. The conformation of the H4 tail differs from that observed in structures of the free nucleosome where the tail makes inter-nucleosomal contacts with the 'acidic patch' of a neighboring nucleosome. It also differs from the H4 position observed in a higher order structure where the H4 tail extends over the DNA interface between two nucleosomes (*Schalch et al., 2005*). A loop in lobe 2 of the ATPase (CHD4 residues 1001–1006) replaces the H4 tail in this position, apparently inducing H4 positioning that allows ATPase lobe 2 binding (*Figure 4a*).

ATPase lobe 2 contains a highly acidic cavity formed by Asp1080, Glu1083, Asp1084, and Glu1087 (*Figure 4a*). This acidic cavity is conserved across all CHD family members. The basic side chain of the H4 histone tail residue Arg17 inserts into this acidic cavity (*Figure 4a*). Similar interactions with the H4 tail have also been reported for Snf2 and ISWI remodelers (*Armache et al., 2019*; *Yan et al., 2019*). The side chain of H4 Lys16 also points toward the acidic cavity and is positioned in close proximity to residues Asp1080 and Glu1083. Acetylation of H4 Lys16 is therefore predicted to weaken these charge-based interactions and to reduce the affinity of chromatin remodellers for the H4 tail. This was noted before (*Yan et al., 2016*) and is consistent with CHD4 activity in repressed regions that lack such H4 acetylation.

## CHD4 interacts with histone H3

The ATPase lobe 2 also contacts the core of histone H3 (alpha helix 1, Gln76 and Arg83) via CHD4 residues Asn1004 and Leu1009, respectively (*Figure 4b*). This contact is critical for chromatin remodeling. Deletion of the homologous region in Chd1 leads to abolishment of chromatin remodeling activity (*Sundaramoorthy et al., 2018*). However, it remains unclear if these contacts are required for proper substrate recognition and positioning or whether they are also necessary to generate the force required for DNA translocation. Low-pass filtering of our map further shows the H3 N-terminal tail trajectory, which extends to the double chromodomain (*Figure 4c*). The contact between the H3 tail and the double chromodomain could target CHD4 to nucleosomes methylated at Lys27 of H3 (*Kuzmichev et al., 2002*), a classical mark for gene repression.

## Two CHD4 molecules can engage with the nucleosome

During 3D classification of our cryo-EM dataset we observed a distinct class of particles that showed two CHD4 molecules bound to the same nucleosome (*Figure 5*, *Figure 1—figure supplements 2–4*, *Video 2*). Refinement of this class of particles yielded a reconstruction at an overall resolution of 4.0 Å (FSC 0.143 criterion) (*Table 1*). A model of this nucleosome-CHD4$_2$ complex was obtained by docking the refined nucleosome-CHD4 model into the density and then placing another CHD4

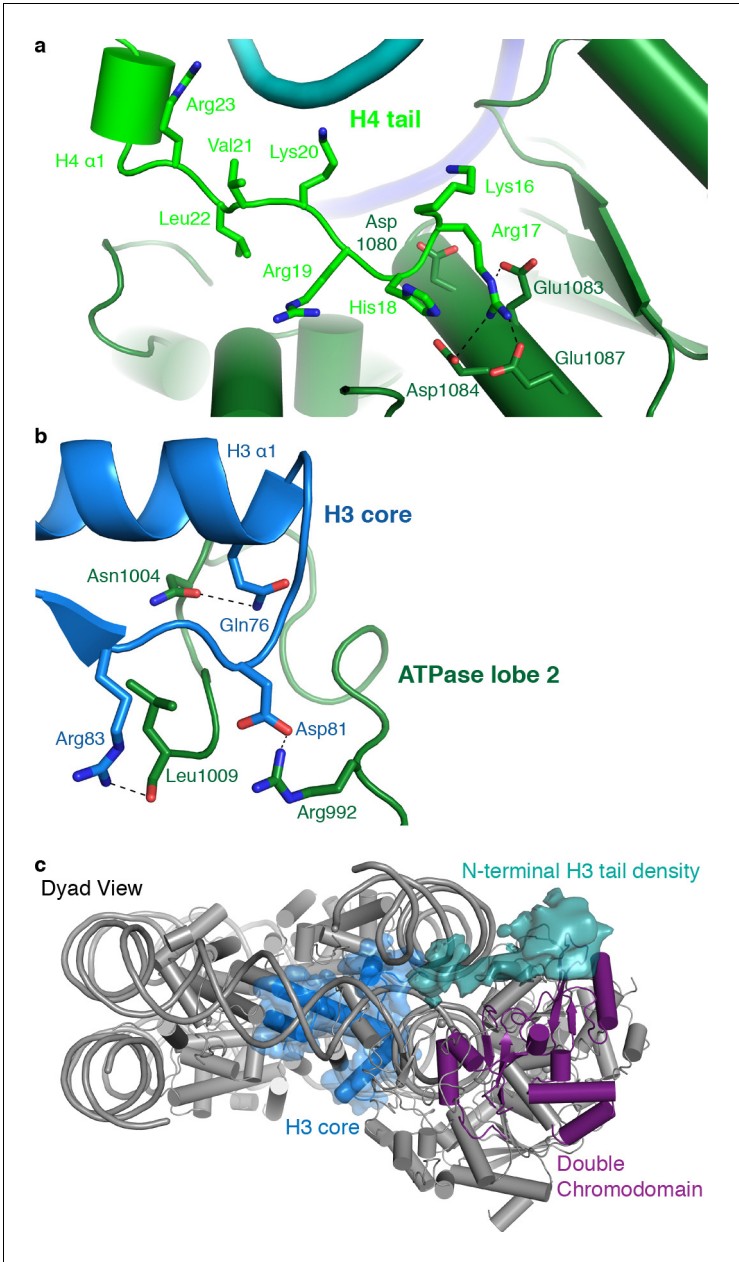

**Figure 4.** CHD4 contacts H3 and H4. (**a**) ATPase lobe 2 interacts extensively with the H4 tail. (**b**) A loop in ATPase lobe 2 contacts H3 alpha helix 1 and neighboring residues. (**c**) The double chromodomain of CHD4 contacts the H3 N-terminal tail. H3 core is shown in blue, H3 tail density from the low-pass filtered final map (7 Å) in teal, and the double chromodomain in purple.

molecule into the additional density observed on the opposite side. The resulting nucleosome-CHD4$_2$ complex structure shows pseudo-twofold symmetry with CHD4 molecules bound at SHL +2 and SHL −2 (*Figure 5*). The second CHD4 molecule uses its double chromodomain and PHD finger 2 to contact nucleosomal DNA at SHL +1 and +0.5, respectively. Binding of the second CHD4 molecule also did not lead to unwrapping of terminal DNA.

Binding of two chromatin remodellers to a single nucleosome was previously observed for *S. cerevisiae* Chd1 (*Sundaramoorthy et al., 2018*) and *H. sapiens* SNF2H (*Armache et al., 2019*). However, in contrast to the structure of the nucleosome-SNF2H$_2$ complex, we do not observe a distortion in the histone octamer due to the presence of the chromatin remodellers. Binding of two remodeler molecules could allow for higher efficiency in positioning the nucleosome at a precise

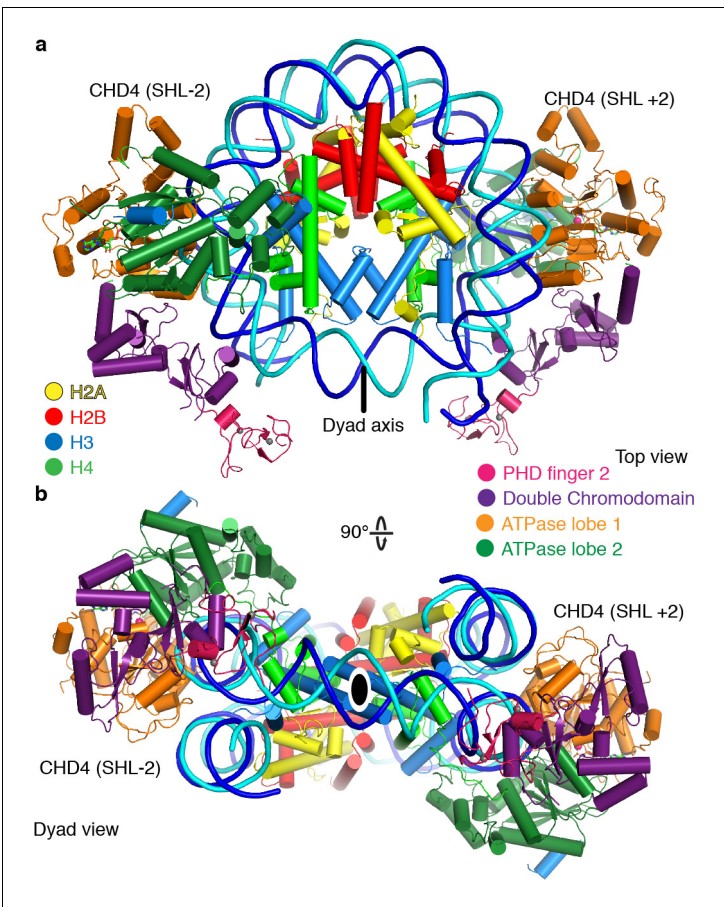

**Figure 5.** The nucleosome can bind two copies of CHD4. Cartoon model of the nucleosome-CHD4$_2$ structure viewed from the top (**a**), and dyad view (**b**).

location but necessitates coordination of the remodellers. A possible mechanism for coordination could be that twist defects that are introduced by remodeler binding are propagated from the entry SHL 2 into the exit side SHL 2 (*Brandani et al., 2018*; *Brandani and Takada, 2018*). Presence of the twist defect at the second remodeler binding site could interfere with the translocation activity of the second remodeler (*Sabantsev et al., 2019*).

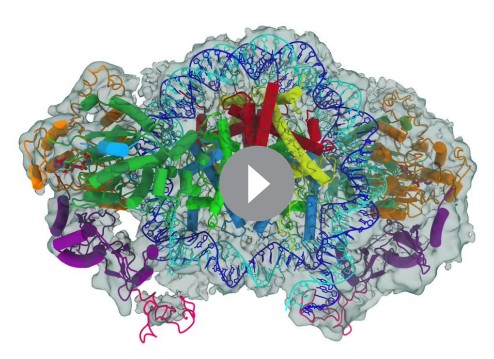

**Video 2.** Cryo-EM density and structure of the nucleosome-CHD4$_2$ complex.
https://elifesciences.org/articles/56178#video2

## Cancer-related CHD4 mutations

Many studies have reported mutations in CHD4 that are related to human diseases, in particular cancer (*Xia et al., 2017*). Mutations involved in various cancer phenotypes have been observed in the PHD finger 2, the double chromodomain, and both lobes of the ATPase motor. To elucidate effects of such mutations on CHD4 activity, the *Drosophila melanogaster* CHD4 homologue Mi-2 has been used as a model protein for functional analysis (*Kovač et al., 2018*). CHD4 mutations have been found to fall in two categories. Whereas some mutations influence ATPase and DNA translocation activity (Arg1162, His1196, His1151 and Leu1215), other mutations seem to change protein stability (Leu912, and Cys464) or disrupt DNA binding (Val558 and Arg572).

To try and rationalize these findings, we mapped known CHD4 mutations on our high-resolution structure (*Figure 6*, *Table 2*). Selected sites of mutation are described below. Mutation of residue His1151 to arginine results in a significant reduction of ATPase activity and abolishes chromatin remodeling activity (*Kovač et al., 2018*). The close proximity of this residue to motif Va (CHD4 residues 1147–1150) makes it likely that the mutation disrupts motif Va function, leading to an uncoupling of the ATPase activity from chromatin remodeling. Similar findings were made for Snf2 where mutation of the tryptophan residue in motif Va resulted in an uncoupling phenotype (*Liu et al., 2017*). The most frequently mutated residue in endometrial cancer, arginine 1162, is located in the ATPase motif VI. It forms an 'arginine finger' that directly interacts with AMP-PNP in our structure. Consistent with this observation, mutation of Arg1162 to glutamine impairs ATP hydrolysis in biochemical assays (*Kovač et al., 2018*).

## Other disease-related CHD4 mutations

De novo missense mutations in CHD4 are also associated with an intellectual disability syndrome with distinctive dysmorphisms (*Sifrim et al., 2016*; *Weiss et al., 2016*). Mutations observed in patients with this syndrome are located in PHD finger 2 (Cys467Tyr) and predominantly in ATPase lobe 2 (Ser851Tyr, Gly1003Asp, Arg1068His, Arg1127Gln, Trp1148Leu, Arg1173Leu, and Val1608Ile). We mapped the sites of these mutations onto our structure (*Figure 6*) and attempted to predict the effects of the mutations as far as possible (*Table 2*).

The Cys467Tyr mutation disrupts coordination of a zinc ion in PHD finger 2. Gly1003 in ATPase lobe2 is located in a loop near H3 alpha helix 1. Deletion of this loop in Chd1 results in a loss of chromatin remodeling activity (*Sundaramoorthy et al., 2018*). Residue Arg1068 forms a hydrogen bond network with the side chain of Thr1137 and the main chain carbonyl groups of Phe1112 and Gln1119. The Arg1068Cys mutation disrupts this network and is predicted to impair the integrity of the ATPase fold. Mutation of Arg1127 disrupts its interactions with the DNA minor groove (*Figure 3c*). The equivalent arginine residue in SMARCA4, which is one of the catalytic subunits of the BAF complex, has been implicated in the rare genetic disorder Coffin-Siris syndrome (*Tsurusaki et al., 2012*). Trp1148, which is part of ATPase motif Va, contacts the guide strand in a fashion similar to Chd1 and Snf2 (*Farnung et al., 2017*; *Liu et al., 2017*; *Figure 3c*). Mutation of this residue uncouples ATP hydrolysis and chromatin remodelling (*Liu et al., 2017*). Arg1173 inserts into an acidic pocket formed by residues Glu971, Asp1147, and Asp1153. Mutation of the arginine residue to leucine is likely to destabilize ATPase lobe 2 folding.

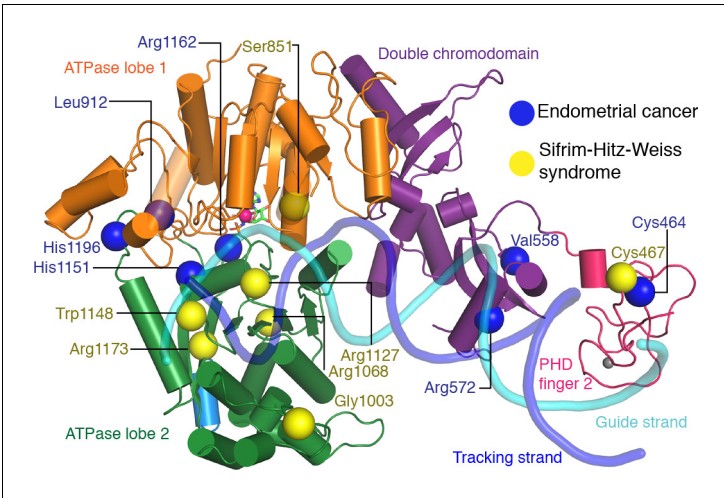

**Figure 6.** CHD4 mutations in cancer and Sifrim-Hitz-Weiss syndrome. Missense mutations that occur in endometrial cancer (blue spheres) and Sifrim-Hitz-Weiss syndrome (yellow spheres) mapped onto the CHD4 structure. Residue numbering is indicated. Nucleosomal DNA at SHL +2 is shown in a semi-transparent cartoon representation.

**Table 2.** CHD4 mutations in cancer and Sifrim-Hitz-Weiss syndrome.

| | Mutated Residue | Location | Predicted effect based on structure | Biochemical observations |
|---|---|---|---|---|
| Cancer | | | | |
| | Cys464Tyr | PHD finger 2 | Disruption of $Zn^{2+}$ binding in PHD finger 2 | Reduction in ATPase activity (***Kovač et al., 2018***) |
| | Val558Phe | Double chromodomain | | Reduced ATPase activity (***Kovač et al., 2018***) |
| | Arg572Gln | Double chromodomain | Disruption of contact with tracking strand | Reduced DNA binding affinity, Loss of full remodeling activity and ATPase activity (***Kovač et al., 2018***) |
| | Leu912Val | ATPase lobe 2 | No prediction made | Reduction of ATPase activity (***Kovač et al., 2018***) |
| | His1151Arg | ATPase lobe 2 | In close proximity to motif Va, might disrupt contact of Trp1148 | Reduction of ATPase activity, abolishment of remodeling activity (***Kovač et al., 2018***) |
| | Arg1162Gln | ATPase lobe 2, motif VI | Located in ATPase motif VI (arginine finger), Disruption of interaction with ATP | Reduction of ATPase activity (***Kovač et al., 2018***) |
| | His1196Tyr | ATPase lobe 2 | Located in the C-terminal bridge region, Removes negative regulation | Speed of chromatin remodeling is increased and better nucleosome centering capability (***Kovač et al., 2018***) |
| | Leu1215 | ATPase lobe 2/ C-terminal bridge | Not located in modeled region | |
| Sifrim-Hitz-Weiss syndrome (***Sifrim et al., 2016***; ***Weiss et al., 2016***) | | | | |
| | Cys467Tyr | PHD finger 2 | Disruption of $Zn^{2+}$ binding in PHD finger 2 | |
| | Ser851Tyr | ATPase lobe 1 | | |
| | Gly1003Asp | ATPase lobe 2 | Disruption of contact with H3 | |
| | Arg1068His | ATPase lobe 2 | Disruption of structural integrity of RecA fold | |
| | Arg1127Gln | ATPase lobe 2 | Disruption of contact with DNA minor groove, equivalent arginine residue in SMARCA4 is implicated in 'Coffin Siris syndrome' | |
| | Trp1148Leu | ATPase lobe 2, motif Va | Disruption of contact with guide strand | Uncoupling of ATPase activity and chromatin remodeling (***Liu et al., 2017***) |
| | Arg1173Leu | | Destabilization | |
| | Val1608Ile | | Not located in modeled region | |

## Discussion

Here, we provide the 3.1 Å resolution cryo-EM structure of human CHD4 engaged with a nucleosome and the 4.0 Å resolution structure of a nucleosome-$CHD4_2$ complex that contains two molecules of CHD4. Our structure of the nucleosome-CHD4 complex reveals how a subfamily II CHD remodeler engages with its nucleosomal substrate. We observe a distortion of nucleosomal DNA at SHL +2 in the presence of AMP-PNP. Similar observations were previously made for the Snf2 chromatin remodeler (***Li et al., 2019***; ***Liu et al., 2017***) in its apo and ADP-bound states.

Our high-resolution structure fills a gap in our understanding of the mechanism of chromatin remodeling by capturing an additional enzymatic state. The DNA distortion at SHL +2 that we observed in the AMP-PNP bound state differs from distortions observed previously in the apo and ADP-bound state that involved a twist distortion (***Li et al., 2019***; ***Winger et al., 2018***). This is consistent with a proposed 'twist defect' mechanism for chromatin remodeling (***Li et al., 2019***; ***Sabantsev et al., 2019***). In this model, binding of the ATPase motor at SHL ± 2 induces a twist defect in the DNA. Subsequent ATP binding, captured by AMP-PNP and ADP·$BeF_3$ structures, then leads to closing of the ATPase motor and to propagation of the twist defect toward the dyad. It is possible that previous nucleosome-Chd1 structures with ADP·$BeF_3$ (***Farnung et al., 2017***;

*Sundaramoorthy et al., 2018*) contained the same DNA distortion but that the lower resolution prevented its observation. Finally, ATP hydrolysis would reset the remodeller and the enzymatic cycle can resume at the next DNA position.

A major difference between the subfamily I remodeller Chd1 and the subfamily II remodeller CHD4 is that Chd1 induces unwrapping of the terminal nucleosomal DNA, whereas CHD4 does not change the DNA trajectory between SHL −7 and −5. DNA unwrapping is observed for Chd1 in structures and in solution and is independent of which ATP or transition state analogue is bound to the motor domain, indicating it is achieved with the use of binding energy only. Our observations are consistent with a single-molecule FRET study (*Zhong et al., 2019*). This major difference in Chd1 and CHD4 molecular function is likely related to a striking difference in cellular function. Whereas Chd1 functions in euchromatic regions of the genome during active transcription (*Skene et al., 2014*), CHD4 plays a central role in the establishment and maintenance of repressive genome regions. Consistent with these findings, DNA unwrapping should be prevented in stable heterochromatic regions. It is possible that these differences in functionality were achieved during evolution by the addition of distinct auxiliary domains in different CHD subfamilies.

Our structure also maps causative disease mutations and helps to investigate how these can impair CHD4 function. Our structure suggests that various mutations may disrupt DNA binding, impede ATP hydrolysis, or uncouple ATP hydrolysis and DNA translocation. The structure thus suggests the effects of CHD4 mutations in cancer and intellectual disability syndromes on chromatin remodeling. It also helps in understanding disease phenotypes of other chromatin remodelers such as the BAF complex that shows a related domain architecture for its ATPase motor. Due to its high resolution, the structure may also guide drug discovery using chromatin remodelers as targets in the future.

## Materials and methods

### Preparation of CHD4

*H. sapiens* CHD4 (Uniprot Accession code Q14839-1) was amplified from human cDNA using the following ligation-independent cloning (LIC) compatible primer pair (Forward primer: 5'-TAC TTC CAA TCC AAT GCA ATG GCG TCG GGC CTG-3', reverse primer: 5'-TTA TCC ACT TCC AAT GTT ATT ACT GCT GCT GGG CTA CCT G-3'). The PCR product containing CHD4 was cloned into a modified pFastBac vector (a gift from S. Gradia, UC Berkeley, vector 438 C, Addgene: 55220) via LIC. The CHD4 construct contains an N-terminal 6xHis tag, followed by an MBP tag, a 10x Asn linker sequence, and a tobacco etch virus protease cleavage site. All sequences were verified by Sanger sequencing.

The CHD4 plasmid (500 ng) was electroporated into DH10EMBacY cells (Geneva Biotech) to generate a bacmid encoding full-length *H. sapiens* CHD4. Bacmids were subsequently selected and prepared from positive clones using blue/white selection and isopropanol precipitation. V0 and V1 virus production was performed as previously described (*Farnung et al., 2017*). Hi5 cells (600 ml) grown in ESF-921 media (Expression Systems) were infected with 200 µl of V1 virus for protein expression. The cells were grown for 72 hr at 27°C. Cells were harvested by centrifugation (238 g, 4°C, 30 min) and resuspended in lysis buffer (300 mM NaCl, 20 mM Na·HEPES pH 7.4, 10% (v/v) glycerol, 1 mM DTT, 30 mM imidazole pH 8.0, 0.284 µg ml$^{-1}$ leupeptin, 1.37 µg ml$^{-1}$ pepstatin A, 0.17 mg ml$^{-1}$ PMSF, 0.33 mg ml$^{-1}$ benzamidine). The cell resuspension was frozen and stored at −80°C.

*H. sapiens* CHD4 was purified at 4°C. Frozen cell pellets were thawed and lysed by sonication. Lysates were cleared by two centrifugation steps (18,000 g, 4°C, 30 min and 235,000 g, 4°C, 60 min). The supernatant containing CHD4 was filtered using 0.8 µm syringe filters (Millipore). The filtered sample was applied onto a GE HisTrap HP 5 ml (GE Healthcare), pre-equilibrated in lysis buffer. After sample application, the column was washed with 10 CV lysis buffer, 5 CV high-salt buffer (1 M NaCl, 20 mM Na·HEPES pH 7.4, 10% (v/v) glycerol, 1 mM DTT, 30 mM imidazole pH 8.0, 0.284 µg ml−1 leupeptin, 1.37 µg ml−1 pepstatin A, 0.17 mg ml−1 PMSF, 0.33 mg ml−1 benzamidine), and 5 CV lysis buffer. The protein was eluted with a gradient of 0–100% elution buffer (300 mM NaCl, 20 mM Na·HEPES pH 7.4, 10% (v/v) glycerol, 1 mM DTT, 500 mM imidazole pH 8.0, 0.284 µg ml$^{-1}$ leupeptin, 1.37 µg ml$^{-1}$ pepstatin A, 0.17 mg ml$^{-1}$ PMSF, 0.33 mg ml$^{-1}$ benzamidine). Peak fractions were pooled and dialysed for 16 hr against 600 ml dialysis buffer (300 mM NaCl, 20 mM Na·HEPES

pH 7.4, 10% (v/v) glycerol, 1 mM DTT, 30 mM imidazole) in the presence of 2 mg His6-TEV protease. The dialysed sample was applied to a GE HisTrap HP 5 ml. The flow-through containing CHD4 was concentrated using an Amicon Millipore 15 ml 50,000 MWCO centrifugal concentrator. The concentrated CHD4 sample was applied to a GE S200 16/600 pg size exclusion column, pre-equilibrated in gel filtration buffer (300 mM NaCl, 20 mM Na·HEPES pH 7.4, 10% (v/v) glycerol, 1 mM DTT). Peak fractions were concentrated to ~40 µM, aliquoted, flash frozen, and stored at −80℃. Typical yields of *H. sapiens* CHD4 from 1.2 L of Hi5 insect cell culture are 2–4 mg.

## Preparation of CHD1

*S. cerevisiae* Chd1 (residues 1–1247) used for FRET assays was cloned, expressed, and purified similarly to the previously described strategy for full-length Chd1 (*Farnung et al., 2017*).

## Nucleosome preparation

*Xenopus laevis* histones were expressed and purified as described (*Dyer et al., 2003*; *Farnung et al., 2017*). DNA fragments for nucleosome reconstitution were generated by PCR essentially as described (*Farnung et al., 2018*). A vector containing the Widom 601 sequence was used as a template for PCR. Super-helical locations are assigned based on previous publications (*Farnung et al., 2018*; *Farnung et al., 2017*; *Kujirai et al., 2018*; *Sundaramoorthy et al., 2018*), assuming potential direction of transcription from negative to positive SHLs. Large-scale PCR reactions were performed with two PCR primers (Structural studies: forward primer: CC TGT TAT TCC TAG TAA TCA ATC AGT GCC TAT CGA TGT ATA TAT CTG ACA CGT GCC T, reverse primer: CCC CAT CAG AAT CCC GGT GCC G; FRET assay: forward primer:/5Cy3/CAA TCA GTG CCT ATC GAT GTA TAT ATC TGA CAC GTG CCT, reverse primer:/5Cy5/CCC CAT CAG AAT CCC GGT GCC G) at a scale of 25 mL. The DNA construct used for structural studies was designed based on previously reported constructs used for the study of CHD remodelers. Nucleosome core particle reconstitution was performed using the salt-gradient dialysis method (*Dyer et al., 2003*). Quantification of the reconstituted nucleosome was achieved by measuring absorbance at 280 nm. Molar extinction coefficients were determined for protein and nucleic acid components and were summed to yield a molar extinction coefficient for the reconstituted extended nucleosome.

## Reconstitution of nucleosome-CHD4 complex

Reconstituted nucleosome core particles and CHD4 were mixed at a molar ratio of 1:2. AMP-PNP was added at a final concentration of 1 mM and the sample was incubated for 10 min on ice. After 10 min compensation buffer was added to a final buffer concentration of 30 mM NaCl, 3 mM $MgCl_2$, 20 mM Na·HEPES pH 7.5, 4% (v/v) glycerol, 1 mM DTT. The sample was applied to a Superose 6 Increase 3.2/300 column equilibrated in gel filtration buffer (30 mM NaCl, 3 mM $MgCl_2$, 20 mM Na·HEPES pH 7.5, 5% (v/v) glycerol, 1 mM DTT). The elution was fractionated in 50 µL fractions and peak fractions were analyzed by SDS-PAGE. Relevant fractions containing nucleosome core particle and CHD4 were selected and cross-linked with 0.1% (v/v) glutaraldehyde. The crosslinking reaction was performed for 10 min on ice and subsequently quenched for 10 min using a final concentration of 2 mM lysine and 8 mM aspartate. The sample was transferred to a Slide-A-Lyzer MINI Dialysis Unit 20,000 MWCO (Thermo Scientific), and dialysed for 4 hr against 600 ml dialysis buffer (30 mM NaCl, 3 mM $MgCl_2$, 20 mM Na·HEPES pH 7.4, 20 mM Tris·HCl pH 7.5, 1 mM DTT). The sample was subsequently concentrated using a Vivaspin 500 ultrafiltration centrifugal concentrator (Sartorius) to a final concentration of ~200–300 µM.

## Cryo-EM analysis and image processing

The nucleosome-CHD4 sample was applied to R2/2 gold grids (Quantifoil). The grids were glow-discharged for 100 s before sample application of 2 µl on each side of the grid. The sample was subsequently blotted for 8.5 s (Blot force 5) and vitrified by plunging into liquid ethane with a Vitrobot Mark IV (FEI Company) operated at 4℃ and 100% humidity. Cryo-EM data were acquired on a Titan Krios transmission electron microscope (FEI/Thermo) operated at 300 keV, equipped with a K2 summit direct detector (Gatan) and a GIF Quantum energy filter. Automated data acquisition was carried out using FEI EPU software at a nominal magnification of 130,000 × in nanoprobe EF-TEM mode.

Image stacks of 40 frames were collected in counting mode over 10 s. The dose rate was ~4.3–4.5 $e^-$ per $\text{Å}^2$ per s for a total dose of ~43–45 $e^-$ $\text{Å}^{-2}$. A total of 3904 image stacks were collected.

Micrograph frames were stacked and processed. All micrographs were CTF estimated and motion corrected using Warp (*Tegunov and Cramer, 2018*). Particles were picked using an in-house trained instance of the neural network BoxNet2 of Warp, yielding 650,598 particle positions. Particles were extracted with a box size of $300^2$ pixel and normalized. Image processing was performed with RELION 3.0-beta 2 (*Zivanov et al., 2018*). Using a 30 Å low-pass filtered *ab initio* model generated in cryoSPARC from 1679 particles (*Figure 1—figure supplement 2c*), we performed one round of 3D classification of all particle images with image alignment. One class with defined density for the nucleosome-CHD4 complex was selected for a second round of classification. The second round of classification resulted in two classes with one copy of CHD4 bound to the nucleosome. The respective classes were selected and 3D refined. The refined nucleosome-CHD4 model was subsequently CTF refined and the beam tilt was estimated based on grouping of beam tilt classes according to their exposure positions. The CTF refined particles were submitted to one additional round of masked 3D classification without image alignment. The mask encompassed CHD4. The most occupied class from this classification was subsequently CTF-refined. The final particle reconstruction was obtained from a 3D refinement with a mask that encompasses the entire nucleosome-CHD4 complex.

The nucleosome-CHD4 reconstruction was obtained from 89,623 particles with an overall resolution of 3.1 Å (gold-standard Fourier shell correlation 0.143 criterion). The final map was sharpened with a *B*-factor of $-36$ $\text{Å}^2$. To exclude that the reconstruction could be a mixture of particles with CHD4 bound to either SHL –2 or SHL +2, CHD4 signal was subtracted and prior angular and translational information for every particle was removed. The subtracted particles were then refined against a synthetic nucleosome core particle map lacking CHD4. As expected, the refinement resulted in a reconstruction where only density for the nucleosome core particle was observed. Subsequently, the particle subtraction was reverted and a 3D classification without image alignment against a single class was performed. This 3D classification employed the angular and translational information provided from the subtraction refinement. The resulting reconstruction showed clear density for CHD4 only at SHL +2, and not at SHL −2, giving a clear indication that the final nucleosome-CHD4 reconstruction contains CHD4 bound only at SHL +2 (*Figure 1—figure supplement 4f*). We cannot rule out, however, that our map is still to some extent a mix of CHD4 bound on either side of the nucleosome.

The second round of 3D classification yielded a class with a nucleosome-CHD4$_2$ complex. The particles were subsequently classified and refined. The resulting reconstruction with 40,233 particles had an overall resolution of 4.0 Å (gold-standard Fourier shell correlation 0.143 criterion). The final map was sharpened with a *B*-factor of $-86$ $\text{Å}^2$. Local resolution estimates for both structures were determined using the built-in RELION tool.

## Model building

Crystal structures of the *X. laevis* nucleosome with the Widom 601 sequence (*Vasudevan et al., 2010*) (PDB code 3LZ0) and the double chromodomain of CHD4 (PDB code 4O9I) were placed into the density of the nucleosome-CHD4 complex as rigid bodies using UCSF Chimera. The protein sequence of the ATPase motor of CHD4 (residues 706–1196) was 'one-to-one threaded' using the ATPase motor of *S. cerevisiae* Chd1 (PDB code 5O9G) as a template by employing Phyre2 (*Kelley et al., 2015*). The threaded model was placed into the density as a rigid body using UCSF Chimera (*Goddard et al., 2018*). Additional density belonging to helical extensions and loops present in the ATPase motor region were modeled de novo. The modeled sequence range 1405–1416 is assigned tentatively based on a previously published Chd1 crystal structure (PDB code 3MWY).

The nucleosome structure, double chromodomain structure, and ATPase motor model were adjusted manually in COOT (version 0.9-pre) (*Emsley et al., 2010*). The structure of PHD finger 2 (*Mansfield et al., 2011*) was then manually placed into the remaining, weaker density next to the double chromodomain and rigid-body docked (*Figure 1—figure supplement 3*), assisted by PDB code 6Q3M. Additional structural elements such as the H4 tail, the C-terminal bridge and loop regions of CHD4 were built using COOT. AMP-PNP and a coordinated $Mg^{2+}$ ion were placed into the corresponding density. AMP-PNP was derived from the monomer library in COOT. The high resolution of our reconstruction enabled us to model some DNA-interacting side chains in two

alternative conformations. The complete model was real-space refined in PHENIX (*Afonine et al., 2018*) with global minimization, local rotamer fitting, morphing, and simulated annealing. To model the nucleosome-CHD4$_2$ complex, the CHD4 model was duplicated and the second copy was rigid body docked into the additional density using UCSF ChimeraX (*Goddard et al., 2018*). The resulting structure was real space refined in PHENIX with global minimization, local rotamer fitting, morphing, and simulated annealing.

### Förster resonance energy transfer (FRET) assay

100 nM of NCP with Cy3 and Cy5 5'-terminal DNA ends was incubated with 300 nM *S. cerevisiae* Chd1 (residues 1–1247) or full-length CHD4 and 1 mM ADP·BeF$_3$ or 1 mM AMP-PNP at final reaction conditions of 50 mM NaCl, 3 mM MgCl$_2$, 20 mM Na·HEPES pH 7.4, 0.1 mg/mL BSA, 10% (v/v) glycerol, 1 mM DTT. To increase FRET efficiency, we used a DNA construct that is shortened by 18 bp on the DNA exit side compared to the construct used for the structural studies. The sample was subsequently incubated for 30 min and transferred to 384-well plates. The reaction was then monitored using a fluorescence emission scan from 520 to 740 nm in a Tecan infinite m1000 pro plate reader with an excitation wavelength of 510 nm. All reactions were performed in triplicates in independent experiments. Emission spectra were normalized by total emissions. Averages of the triplicates and corresponding standard deviations are reported. The results were plotted using Matplotlib.

### Figure generation

Figures were generated using PyMol (version 2.2.2) and UCSF ChimeraX.

## Acknowledgements

We thank past and present members of the Cramer laboratory. We thank C Oberthür for help with protein purification, U Neef for insect cell maintenance, A Sawicka for providing cDNA, and SM Vos for valuable input and critical reading of the manuscript. We thank C Dienemann and U Steuerwald for support with electron microscopy. PC was supported by the Deutsche Forschungsgemeinschaft (SFB1064, SFB860), the European Research Council Advanced Investigator Grant TRANSREGULON (grant agreement No. 693023), and the Volkswagen Foundation.

## Additional information

### Funding

| Funder | Grant reference number | Author |
| --- | --- | --- |
| Deutsche Forschungsgemeinschaft | SFB1064 | Patrick Cramer |
| Deutsche Forschungsgemeinschaft | SFB860 | Patrick Cramer |
| European Research Council | 693023 | Patrick Cramer |
| Volkswagen Foundation | | Patrick Cramer |
| Deutsche Forschungsgemeinschaft | EXC 2067/1-390729940 | Patrick Cramer |

The funders had no role in study design, data collection and interpretation, or the decision to submit the work for publication.

### Author contributions

Lucas Farnung, Conceptualization, Formal analysis, Supervision, Validation, Investigation, Visualization, Methodology, Writing - original draft, Writing - review and editing; Moritz Ochmann, Investigation, Visualization, Writing - review and editing; Patrick Cramer, Conceptualization, Resources, Formal analysis, Supervision, Funding acquisition, Investigation, Methodology, Writing - original draft, Project administration, Writing - review and editing

Author ORCIDs

Lucas Farnung (iD) https://orcid.org/0000-0002-8200-2493
Patrick Cramer (iD) https://orcid.org/0000-0001-5454-7755

Decision letter and Author response

Decision letter https://doi.org/10.7554/eLife.56178.sa1
Author response https://doi.org/10.7554/eLife.56178.sa2

## Additional files

### Supplementary files

• Transparent reporting form

### Data availability

The cryo-EM reconstructions and final models were deposited with the Electron Microscopy Data Base (accession codes EMD-10058 and EMD-10059) and with the Protein Data Bank (accession code 6RYR and 6RYU). The raw image data and corresponding WARP sessions have been deposited to EMPIAR (EMPIAR-10411).

The following datasets were generated:

| Author(s) | Year | Dataset title | Dataset URL | Database and Identifier |
|---|---|---|---|---|
| Farnung L, Ochmann M, Cramer P | 2020 | Nucleosome-CHD4 complex structure (single CHD4 copy) | http://www.ebi.ac.uk/pdbe/entry/emdb/EMDB-10058 | Electron Microscopy Data Bank, EMDB-10058 |
| Farnung L, Ochmann M, Cramer P | 2020 | Single Particle Cryo-EM Reconstructions of NCP-CHD4 complexes | http://www.ebi.ac.uk/pdbe/emdb/empiar/entry/10411/ | Electron Microscopy Public Image Archive, EMPIAR-10411 |
| Farnung L, Ochmann M, Cramer P | 2020 | Nucleosome-CHD4 complex structure (two CHD4 copies) | http://www.ebi.ac.uk/pdbe/entry/emdb/EMDB-10059 | Electron Microscopy Data Bank, EMDB-10059 |
| Farnung L, Ochmann M, Cramer P | 2020 | Nucleosome-CHD4 complex structure (single CHD4 copy) | https://www.rcsb.org/structure/6RYR | RCSB Protein Data Bank, 6RYR |
| Farnung L, Ochmann M, Cramer P | 2020 | Nucleosome-CHD4 complex structure (two CHD4 copies) | https://www.rcsb.org/structure/6RYU | RCSB Protein Data Bank, 6RYU |

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
