## [Decision Letter]

**Acceptance summary:**

The dearth of atomic resolution structures has been a major roadblock in understanding the mechanisms of chromatin remodelers. The high-quality EM structure and the associated EM data in this work is expected to be a great resource for the chromatin field to develop and test mechanistic models.

**Decision letter after peer review:**

[Editors’ note: the authors submitted for reconsideration following the decision after peer review. What follows is the decision letter after the first round of review.]

Thank you for submitting your work entitled "Nucleosome-CHD4 chromatin remodeller structure explains human disease mutations" for consideration by *eLife*. Your article has been reviewed by three peer reviewers, one of whom is a member of our Board of Reviewing Editors, and the evaluation has been overseen by a Senior Editor. The reviewers have opted to remain anonymous.

Our decision has been reached after consultation between the reviewers. Based on these discussions and the individual reviews below, we regret to inform you that your work will not be considered for publication in *eLife*.

As you will see from the individual reviews attached below all the reviewers found the structure to be of high quality. However, in the discussion amongst the reviewers it was also agreed that the structure by itself does not sufficiently advance our understanding of remodeling mechanisms. Comparison with previous Chd1 structures forms a key component of the mechanistic conclusions made in this work. The reviewers thought that such direct comparisons were not appropriate primarily because of the different ATP states of the two structures. Additionally, the first Chd1 structure had other proteins in the mix that are not present in the context of CHD4. Comparing structures of CHD4 and Chd1 bound to nucleosomes in the presence of the same ATP analog would be ideal. However, the reviewers recognized that this may be difficult and take time. So, in their discussion they tried to come up with possible biochemical experiments (instead of another structure) that could be carried out to test the mechanistic predictions of your current CHD4-nucleosome structure. These are summarized below.

1) Directly compare terminal DNA unwrapping by CHD4 and Chd1 (positive control) in the presence of AMP-PNP and ADP-BeF_x_ using a FRET assay.

2) Introduce gaps at SHL 2 and determine if deformation of the DNA at SHL 2 has a functional role.

3) Test the functional effects of the new contacts and compare to corresponding effects with Chd1.

If any of these biochemical experiments or any others that you choose to carry out provide new mechanistic insights, then we would welcome a resubmission.

Reviewer #1:

Recent high-resolution cryo-EM structures of ATP-dependent chromatin remodelers are shedding new light on the mechanisms of these complex enzymes. Here the authors use cryo-EM to study the structure of CHD4, a human CHD enzyme, bound to a nucleosome in the presence of AMP-PNP. They find that the ATPase domain binds at SHL 2 as predicted from studies of other CHD enzymes and observe a distortion of nucleosomal DNA at this location. However, in contrast to their earlier structure of *S. cerevisiae* Chd1, the authors do not see unpeeling of the terminal DNA that is proximal to the bound ATPase domain. They argue that this difference may arise because CHD4 does not have the SANT-SLIDE DNA binding domains present in Chd1. They also observe interactions with the H4 tail as predicted by previous studies. Finally, they map specific disease associated mutations map on to the ATPase and accessory domains of Chd4.

Overall, the cryo-EM data appears to be of very high quality. My main concern is that the findings do not substantially move the field forward compared to previous structural studies on chromatin remodelers. There are some new findings in this work that have the potential to be mechanistically significant, but these need additional tests. Below are some suggestions for raising the mechanistic impact.

1) The authors suggest that the absence of unpeeled DNA compared to Chd1 may arise because of the absence of a known DNA binding domain. This difference could in principle point to significant mechanistic differences between the two CHD proteins due to different ways of engaging the DNA. However, the differences could also arise due to the difference in nucleotide state between the two studies. The author's previous study used ADP-BeF_x_ while this study used AMP-PNP. To test for mechanistic differences between Chd4 and Cdh1 the authors should compare the two structures in the same ATP state.

2) The presence of distorted DNA is interesting, but the extent of distortion appears much less than observed with Snf2. The authors should test the mechanistic significance of the distortion by classical approaches previously used with ISWI and SWI/SNF enzymes. These experiments used nucleosomes with nicks or gaps at SHL 2. If the authors find substantial reduction in nucleosome sliding but not much reduction in nucleosome binding using nucleosomes with gaps or nicks at SHL 2, then this will imply that distortion of the DNA at SHL 2 plays a functional role.

3) The authors should test the consequences of mutating the specific interactions identified in their structure on remodeling by CHD4. This includes interactions made with the H4 tail, the histone surfaces and DNA. Comparison of the magnitude of the effects of these mutations with the corresponding published effects observed with Snf2, Chd1 and ISWI enzymes will allow for a direct comparison of how different remodelers use interactions with the nucleosome to drive octamer sliding.

Reviewer #2:

ATP-dependent chromatin remodeling is one of the mechanisms to modulate nucleosome structure, using the energy from ATP hydrolysis and dedicated ATPases. These macromolecular machines are employed and influence a range of processes, such as transcription, replication or repair. To maintain or effectuate changes in chromatin, eukaryotic cells evolved four known major families of remodeling factors: SWI/SNF, ISWI, Ino80 and CHD, that have their unique properties.

Farnung et al. report a high-resolution structure of CHD4 bound to a nucleosome, solved using cryo-EM. In their article, they use a full-length human CHD4 construct and a nucleosome with additional extranucleosomal DNA (4 bp on the entry and 30 bp at the exit of the nucleosome) and reveal the mode of binding and interaction between them. This is the first structure of group II CHD ATP-dependent remodelers, following on the studies by Farnung et al., 2017 of group I remodeler, yeast CHD1.

This is a well-written, strictly structural paper that provides a view on an important chromatin remodeler that in the eukaryotic cells is a part of larger remodeling complexes, such as NuRD and ChAHP. The authors could have provided some FRET data to confirm their model. This is important, especially since they derive functional differences from structural comparison of two not-fully resolved remodelers from different organisms on different nucleosomal substrates, using different ATP analogues. However, after some modifications, this data and paper could be accepted, as it is an important and well-approached study.

1) Upon comparing the PDB 3LZ0 with 6RYR, authors modeled 4 base-pairs on the exit site, which leads to the following question:

a) The map EMD-10058_6RYR: If there is 4bp on the entry, and 30bp at the exit, upon filtering the density, why would there be such a large additional amount of density on the entry-side at the low threshold level? Could that be an indication of a mixed population in the final reconstruction, or a translocation?

2) The authors write: "Structural comparisons show that CHD4, in contrast to Chd1, does not induce unwrapping of terminal DNA." The follow up to this is: "In contrast to the nucleosome-Chd1 structure (Farnung et al., 2017), we did not observe unwrapping of nucleosomal DNA from the histone octamer on the second DNA gyre at SHL -6 and -7 (Figure 2)". However, upon closer examination, this DNA density, while not unwrapped, appears to exhibit a lower occupancy than the rest of the nucleosome. Is that an effect of incomplete classification, or a partial distortion by the remodeler?

3) Regarding CHD1 and CHD4 comparison:

a) If comparing it only to the CHD1 publication from 2017 by Farnung et al., the sample there was significantly different. It involved CHD1-FACT-Paf1C-nucleosome assembly, of which only CHD1-nucleosome was visible. The DNA substrate was also different, containing 63 base pairs of extranucleosomal DNA on the exit side vs 30 in this study. Sundaramoorthy et al., 2018 contained only CHD1-nucleosome, but also with a different nucleosome substrate. Could the extranucleosomal DNA be of significance in this study, and could the authors provide a 1-2 sentence explanation of their particular exit/entry side DNA construct rationale?

b) Authors are comparing a yeast CHD1 bound by ADP-BeF_x_3, with human CHD4 bound by AMP-PNP. However, there were reported published cases where a comparison of states using cryo-EM involving those two ATP-analogues yielded different results, or different distribution of states (e.g.: https://www.ncbi.nlm.nih.gov/pmc/articles/PMC5307450/). Could it be completely excluded here?

Reviewer #3:

This manuscript describes a high-resolution cryo-EM structure of CHD4 bound to a nucleosome, thus adding to a small but growing number of chromatin remodeling protein structures. The main strength of the work is the high-resolution structure of this disease-relevant protein and the ability to discuss previous work in a structural context. Comparisons are made to previously defined remodeler-nucleosome structures, which mainly point to commonalities with a few noted exceptions including a lack of exit-side DNA distortion. An intriguing minor deformation in DNA trajectory is observed in the presence of AMP-PNP, which is made possible by the high resolution. Other strengths include determination of the placement of double chromodomains and one of two PHD fingers (albeit to lower resolution) as well as determination of a CHD3/4/5-specific contact with nucleosomal DNA. CHD4 disease-specific mutations are mapped onto the structure to help explain their previously defined functional consequences.

The biggest limitation of the work is the lack of functional information. This manuscript is entirely structural modeling. Though an exceptionally high-resolution structure for a remodeler-nucleosome complex, many of the findings were previously predicted, which makes the insight from the structure more limited than other contemporary remodeler-nucleosome structures. Another limitation is that the structure lacks a significant portion of the CHD4 protein, though this is not mentioned explicitly in the text. Figure 1 displays the fragment that was modeled, but it would be worthwhile to discuss that regions of CHD4 that may be functional or contributing to the remodeling cycle are not resolved. Otherwise it would be important to show that the modeled structure that is being used to justify a remodeling mechanism is sufficient for CHD4 remodeling.

It would be useful to incorporate at least some functional data, particularly to demonstrate function of the newly defined contacts (e.g. loop 832-837, Asn1010, Arg1068, Arg1173) to help solidify structure-function relationships that are proposed.

The title contains "explains disease mutations" but most (perhaps all) of the functional consequences of these disease mutations were defined previously in Mi-2 (Kovac et al) or could have been predicted from highly-conserved ATPase regions. While the manuscript gives credit to the previous characterizations, I believe the title overstates the contribution of the structure. Even simply changing to "Nucleosome-CHD4 chromatin remodeller structure maps human disease mutations" would be more acceptable. In addition, making it transparent in the Abstract that the effects of these mutations were previously defined would be appreciated.

Similarly, the Discussion states that "our structure elucidates the mechanism of chromatin remodeling". More appropriately, the structure may add to or fill gaps in models of the chromatin remodeling mechanism.

The exit-side Discussion section relating heterochromatin to lack of exit-side DNA dynamics in the cryo-EM structure seems speculative and unsupported. It is simpler to state that the lack of unwrapping at the exit side may be due to lack of DNA binding domain. However, unwrapping may still be seen in the context of CHD4 complexes like NuRD and ChAHP. It may be too early to speculate about intermediate DNA unwrapping states, as this structure captures one part of the CHD4 remodeling cycle that is complex and may require some distortion of DNA on both sides for nucleosome repositioning to occur.

[Editors’ note: further revisions were suggested prior to acceptance, as described below.]

Thank you for submitting your work entitled "Nucleosome-CHD4 chromatin remodeller structure maps human disease mutations" for consideration by *eLife*. Your article has been reviewed by three peer reviewers, one of whom is a member of our Board of Reviewing Editors, and the evaluation has been overseen by a Senior Editor. The reviewers have opted to remain anonymous.

Earlier we had said that if any of the biochemical experiments that we had listed or any others that you choose to carry out provide new mechanistic insights then we would welcome a resubmission. In this context, the reviewers appreciated the addition of biochemical data to the resubmission. However, after extensive discussion all reviewers felt that the added FRET data did not provide sufficient new mechanistic insight to merit publication as a Research Article. At the same time, they agreed that your high-quality EM structure and the EM data would be a valuable resource for the chromatin field to develop and test mechanistic models. We can therefore offer to consider your work under the Tools and Resources category of *eLife*.

If you would like for your paper to be published in the Tools and Resources category, we ask that you submit a revised version that addresses the remaining EM-related questions. The major EM related question is summarized below.

As pointed out by our EM expert reviewer it appears that the structure could be a mix of the SHL +2 and SHL -2 bound particles. In this situation, the way to clearly define the structure would be to mask out or delete computationally the density for CHD4 and align particles based on some other feature. Then, without again aligning the data, go back to the original particles and project back the resulting information. This could show if there are SHL +2 and SHL -2 bound particles – the final maps could be at lower resolutions, nevertheless, it would be more precise. Even if this analysis shows no mixture of particles, we feel the analysis is necessary for conclusive interpretation of the data.

Carrying out such an analysis would comprehensively address the following two prior questions asked by the reviewer.

1) Upon comparing the PDB 3LZ0 with 6RYR, authors modeled 4 base-pairs on the exit site, which leads to the following question: The map EMD-10058_6RYR: If there is 4bp on the entry, and 30bp at the exit, upon filtering the density, why would there be such a large additional amount of density on the entry-side at the low threshold level? Could that be an indication of a mixed population in the final reconstruction, or a translocation?

2) The authors write: "Structural comparisons show that CHD4, in contrast to Chd1, does not induce unwrapping of terminal DNA." The follow up to this is: "In contrast to the nucleosome-Chd1 structure (Farnung et al., 2017), we did not observe unwrapping of nucleosomal DNA from the histone octamer on the second DNA gyre at SHL -6 and -7 (Figure 2)". However, upon closer examination, this DNA density, while not unwrapped, appears to exhibit a lower occupancy than the rest of the nucleosome. Is that an effect of incomplete classification, or a partial distortion by the remodeler?

Reviewer #1:

The FRET data presented by the authors addresses part of my core concern about comparing DNA unpeeling by Chd4 vs. Chd1. However, the experiments are quite minimal. Beyond suggesting some unpeeling in one context and not in the other, they do not provide much new insight into the mechanistic differences between Chd1 and Chd4.

In addition, several controls are missing.

1) The authors need to mention describe how they ensured that the nucleosome was fully bound by Chd1 and Chd4 under the FRET experimental conditions. This is important in the case of Chd4, which shows no FRET change.

2) To rule out any environmental effects on the Cy dyes upon binding by Chd1, the authors should show a scan for direct excitation of Cy5. The absence of an effect with direct excitation of Cy5 would rule out an environmental effect on Cy5.

3) It is not clear what is the significance of the extent of FRET change. Some comparison to the FRET changes observed upon increasing salt would allow for calibrating the observed effect with respect to other contexts where DNA is unpeeled.

Reviewer #2:

This revised manuscript from Farnung et al. presents a high resolution cryo-EM structure of the nucleosome core particle in complex with human CHD4, a disease-relevant chromatin remodeling protein with known roles in transcriptional repression. The structure is a nice addition to our limited but growing number of chromatin remodeling proteins in complex with nucleosome substrates. The work defines a potential remodeling intermediate where nucleosomal DNA is distorted at SHL +2, and contrasts CHD4 with Chd1 by showing CHD4 does not unwrap terminal DNA. The structure provides a valuable resource to the chromatin community and it bridges the gap between known structure, known chromatin remodeler function, and known disease mutations.

The major drawback from the work is the limited insight into new chromatin remodeler biology. There is limited new knowledge gained regarding how chromatin remodeling proteins work, and the function of most of the disease mutations were predictable based on a significant number of previous publications describing ATPase and chromatin remodeler biochemistry. Even the described remodeling intermediate is speculative, since there are no comparable structures of CHD4 in other nucleotide-bound states, so the extent of disruption of SHL +2 as a function of nucleotide state is unclear. As mentioned in the previous review, minimal biochemical experiments testing some of the newly identified remodeler-nucleosome contact points would have been quite useful to address these concerns.

Reviewer #3:

ATP-dependent chromatin remodeling is one of the mechanisms to modulate nucleosome structure, using the energy from ATP hydrolysis and dedicated ATPases. These macromolecular machines are employed and influence a range of processes, such as transcription, replication or repair. To maintain or effectuate changes in chromatin, eukaryotic cells evolved four known major families of remodeling factors: SWI/SNF, ISWI, Ino80 and CHD, that have their unique properties.

Farnung et al. report a high-resolution structure of CHD4 bound to a nucleosome, solved using cryo-EM. In their article, they use a full-length human CHD4 construct and a nucleosome with additional extranucleosomal DNA (4 bp on the entry and 30 bp at the exit of the nucleosome) and reveal the mode of binding and interaction between them. This is the first structure of group II CHD ATP-dependent remodelers, following on the studies by Farnung et al., 2017 of group I remodeler, yeast CHD1.

This is a well-written, well-approached structural paper that provides a view on an important chromatin remodeler that in the eukaryotic cells is a part of larger remodeling complexes, such as NuRD and ChAHP.

This is the second time this paper has been reviewed by me; thus, upon reading the paper and the changes made by the authors, I took the liberty of resubmitting this review in an almost unchanged form. From the previous submission, it is clear that the authors took their time to address the most criticized part and performed FRET studies. This is convincing and removes the greatest hurdle in their paper.

However, I still want to request answers to the question I made in the past, that has not been addressed.

I could not see it from the classification, so, could authors provide an indication of how they made sure that their nucleosome orientation in the reconstruction was done properly (30/4bp)? Could it be that the single CHD4 could bind on either side (as in Armache et al., 2019, SNF2H), and thus the EM assignment would have to be extra careful?

[Editors’ note: further revisions were suggested prior to acceptance, as described below.]

Thank you for resubmitting your work entitled "Nucleosome-CHD4 chromatin remodeller structure maps human disease mutations" for further consideration by *eLife*. Your revised article has been evaluated by Jessica Tyler (Senior Editor) and a Reviewing Editor.

The reviewers agree that the work will provide a very valuable resource to the chromatin community. However, our EM expert reviewer has some remaining concerns. Below are excerpted the two key issues that the reviewer raises:

Since the authors decided not to provide a deeper insight into the mixed dataset angle I asked about before, and the density on either side of the nucleosome seems to not clearly support the statement about the 4bp, I would want them to add a sentence about this in the text. A statement, such as, for example: "We cannot rule out that our map is still to some extent a mix of the CHD4 bound on either side of the nucleosome, as filtering of the map suggests the presence of more than 4 bp". As this is to be a resource paper, I also want to suggest the paper acceptance provided that the raw EM data is uploaded to EMPIAR as soon as possible.

All three reviewers have discussed these comments and agree that the following actions are needed prior to acceptance:

1) Addition of the following sentence in the text: "We cannot rule out that our map is still to some extent a mix of the CHD4 bound on either side of the nucleosome, as filtering of the map suggests the presence of more than 4 bp".

2) Uploading the raw EM data to EMPIAR.

---

## [Author Response]

[Editors’ note: the authors submitted for reconsideration following the decision after peer review. What follows is the decision letter after the first round of review.]

[…]1) Directly compare terminal DNA unwrapping by CHD4 and Chd1 (positive control) in the presence of AMP-PNP and ADP-BeF_x_ using a FRET assay.2) Introduce gaps at SHL 2 and determine if deformation of the DNA at SHL 2 has a functional role.3) Test the functional effects of the new contacts and compare to corresponding effects with Chd1.If any of these biochemical experiments or any others that you choose to carry out provide new mechanistic insights, then we would welcome a resubmission.

We thank the reviewers for good suggestions. We now provide the key additional biochemical experiment that the reviewers suggested, the direct comparison of Chd1 and CHD4 in their ability to detach nucleosomal DNA, as monitored by FRET (Suggestion #1). We carried out these assays in the presence of ADP.BeF3 or AMP-PNP. We found that, as expected from the structural data and comparisons, Chd1 detaches/unwraps DNA in the presence of either ATP or transition state analogue, whereas CHD4 does not. This assay complements our structural comparison with Chd1 and indicates an important mechanistic difference between these two CHD proteins. The new data and figure clearly improved our manuscript by providing not only structural but also biochemical evidence for mechanistic differences between members of two CHD subfamilies. We think the resulting manuscript is acceptable for publication. Please also refer to the detailed responses with respect to individual concerns by reviewers below.

Reviewer #1:[…]1) The authors suggest that the absence of unpeeled DNA compared to Chd1 may arise because of the absence of a known DNA binding domain. This difference could in principle point to significant mechanistic differences between the two CHD proteins due to different ways of engaging the DNA. However, the differences could also arise due to the difference in nucleotide state between the two studies. The author's previous study used ADP-BeF_x_ while this study used AMP-PNP. To test for mechanistic differences between Chd4 and Cdh1 the authors should compare the two structures in the same ATP state.

We thank the reviewer for their comment. As explained above, we have addressed this by a FRET assay that monitors DNA unwrapping. The FRET assay was performed with both remodellers in the presence of ADP.BeF_3_- or AMP-PNP. Indeed, Chd1 unwraps DNA in the presence of either ATP/transition state analogue, whereas CHD4 does not. These data are presented now along the structural comparisons of the Chd1- and CHD4-nucleosome complexes in Figure 2B, C. The FRET assay confirms a significant mechanistic difference between these two CHD proteins. With respect to new structures, we trust the reviewer understands this is beyond the scope of the current work, and this is also what we understood from the editorial decision.

2) The presence of distorted DNA is interesting, but the extent of distortion appears much less than observed with Snf2. The authors should test the mechanistic significance of the distortion by classical approaches previously used with ISWI and SWI/SNF enzymes. These experiments used nucleosomes with nicks or gaps at SHL 2. If the authors find substantial reduction in nucleosome sliding but not much reduction in nucleosome binding using nucleosomes with gaps or nicks at SHL 2, then this will imply that distortion of the DNA at SHL 2 plays a functional role.

See above, we did not use this approach but concentrated to the comparisons between CHD4 and Chd1. We trust the reviewer understands this is beyond the scope of the current work, and this is also what we understood from the editorial decision.

3) The authors should test the consequences of mutating the specific interactions identified in their structure on remodeling by CHD4. This includes interactions made with the H4 tail, the histone surfaces and DNA. Comparison of the magnitude of the effects of these mutations with the corresponding published effects observed with Snf2, Chd1 and ISWI enzymes will allow for a direct comparison of how different remodelers use interactions with the nucleosome to drive octamer sliding.

See above, we did not use this approach but concentrated to the comparisons between CHD4 and Chd1. We trust the reviewer understands this is beyond the scope of the current work, and this is also what we understood from the editorial decision.

Reviewer #2:[…]1) Upon comparing the PDB 3LZ0 with 6RYR, authors modeled 4 base-pairs on the exit site, which leads to the following question:a) The map EMD-10058_6RYR: If there is 4bp on the entry, and 30bp at the exit, upon filtering the density, why would there be such a large additional amount of density on the entry-side at the low threshold level? Could that be an indication of a mixed population in the final reconstruction, or a translocation?

We thank the reviewer for their comment. Our analysis revealed additional density of extranucleosomal DNA on the entry and exit side. The entry side, however, only showed additional density for approximately 4 base pairs as can be expected from the used DNA sequence. We did not choose to model these additional 4 base pairs because of the poor signal/noise for this region. In contrast, we were able to observe less ambiguous density at the DNA exit side and accordingly modelled 4 clearly distinguishable base pairs. We would like to point out that a mixed population or translocation events are highly unlikely due to the fact that we were able to assign the DNA register unambiguously using the density around the nucleosomal dyad, which exhibits high resolution (Figure 3C).

2) The authors write: "Structural comparisons show that CHD4, in contrast to Chd1, does not induce unwrapping of terminal DNA." The followup to this is: "In contrast to the nucleosome-Chd1 structure (Farnung et al., 2017), we did not observe unwrapping of nucleosomal DNA from the histone octamer on the second DNA gyre at SHL -6 and -7 (Figure 2)". However, upon closer examination, this DNA density, while not unwrapped, appears to exhibit a lower occupancy than the rest of the nucleosome. Is that an effect of incomplete classification, or a partial distortion by the remodeler?

The lower occupancy is likely due to a higher flexibility at SHL -7 and for the extranucleosomal DNA. Nevertheless, the flexibility around SHL -7 is comparable with the flexibility observed for canonical NCPs (see also FRET assay in Figure 2). Therefore, it is unlikely to be due to a partial distortion by the remodeler.

3) Regarding CHD1 and CHD4 comparison:a) If comparing it only to the CHD1 publication from 2017 by Farnung et al., the sample there was significantly different. It involved CHD1-FACT-Paf1C-nucleosome assembly, of which only CHD1-nucleosome was visible. The DNA substrate was also different, containing 63 base pairs of extranucleosomal DNA on the exit side vs 30 in this study. Sundaramoorthy et al., 2018 contained only CHD1-nucleosome, but also with a different nucleosome substrate. Could the extranucleosomal DNA be of significance in this study, and could the authors provide a 1-2 sentence explanation of their particular exit/entry side DNA construct rationale?

We thank the reviewer for pointing this out. As indicated in Figure 1—figure supplement 1, we used the canonical 145 bp Widom 601 with additional 4 bp on the entry side of the DNA and 30 bp of linker DNA. The full sequence is given in the Materials and methods section of the submitted manuscript. The construct was designed based on the published NCP-Chd1 structures that clearly indicated that a construct with 30 bp of extranucleosomal DNA is sufficient for binding by a DNA binding region (as present in Chd1). We added a corresponding statement on this to the manuscript Materials and methods section.

b) Authors are comparing a yeast CHD1 bound by ADP-BeF_x_3, with human CHD4 bound by AMP-PNP. However, there were reported published cases where a comparison of states using cryo-EM involving those two ATP-analogues yielded different results, or different distribution of states (e.g.: https://www.ncbi.nlm.nih.gov/pmc/articles/PMC5307450/). Could it be completely excluded here?

We have addressed this comment by a FRET assay to monitor DNA unwrapping by both chromatin remodellers in the presence of ADP.BeF3 or AMP-PNP. We found that Chd1 unwraps DNA in the presence of either ATP/transition state analogue, whereas CHD4 does not. Thus, the concern is excluded.

Reviewer #3:[…]The biggest limitation of the work is the lack of functional information. This manuscript is entirely structural modeling. Though an exceptionally high-resolution structure for a remodeler-nucleosome complex, many of the findings were previously predicted, which makes the insight from the structure more limited than other contemporary remodeler-nucleosome structures. Another limitation is that the structure lacks a significant portion of the CHD4 protein, though this is not mentioned explicitly in the text. Figure 1 displays the fragment that was modeled, but it would be worthwhile to discuss that regions of CHD4 that may be functional or contributing to the remodeling cycle are not resolved. Otherwise it would be important to show that the modeled structure that is being used to justify a remodeling mechanism is sufficient for CHD4 remodeling.It would be useful to incorporate at least some functional data, particularly to demonstrate function of the newly defined contacts (e.g. loop 832-837, Asn1010, Arg1068, Arg1173) to help solidify structure-function relationships that are proposed.

We thank the reviewer for this suggestion. Note we have added the key experiment that demonstrates a key difference between CHD4 and Chd1 in their ability to detach/unwrap terminal DNA. Rather than adding details, this experiment and comparison provide new conceptual insights into the CHD family of remodelers. See above for details.

The title contains "explains disease mutations" but most (perhaps all) of the functional consequences of these disease mutations were defined previously in Mi-2 (Kovac et al) or could have been predicted from highly-conserved ATPase regions. While the manuscript gives credit to the previous characterizations I believe the title overstates the contribution of the structure. Even simply changing to "Nucleosome-CHD4 chromatin remodeller structure maps human disease mutations" would be more acceptable. In addition, making it transparent in the Abstract that the effects of these mutations were previously defined would be appreciated.

We thank the reviewer for their suggestion and have changed the title of the manuscript accordingly to “Nucleosome-CHD4 chromatin remodeller structure maps human disease mutations”.

Similarly, the Discussion states that "our structure elucidates the mechanism of chromatin remodeling". More appropriately, the structure may add to or fill gaps in models of the chromatin remodeling mechanism.

We have changed the Discussion accordingly.

The exit-side Discussion section relating heterochromatin to lack of exit-side DNA dynamics in the cryo-EM structure seems speculative and unsupported. It is simpler to state that the lack of unwrapping at the exit side may be due to lack of DNA binding domain. However, unwrapping may still be seen in the context of CHD4 complexes like NuRD and ChAHP. It may be too early to speculate about intermediate DNA unwrapping states, as this structure captures one part of the CHD4 remodeling cycle that is complex and may require some distortion of DNA on both sides for nucleosome repositioning to occur.

We agree and have made sure this is correctly understood. Note we added the key experiment showing DNA dynamics in solution with the use of FRET. See above for details.

[Editors’ note: further revisions were suggested prior to acceptance, as described below.]

If you would like for your paper to be published in the Tools and Resources category, we ask that you submit a revised version that addresses the remaining EM-related questions. The major EM related question is summarized below.

We agree to publish our manuscript in the Tools and Resources category.

We have addressed the remaining EM-related questions as detailed below.

As pointed out by our EM expert reviewer it appears that the structure could be a mix of the SHL +2 and SHL -2 bound particles. In this situation, the way to clearly define the structure would be to mask out or delete computationally the density for CHD4 and align particles based on some other feature. Then, without again aligning the data, go back to the original particles and project back the resulting information. This could show if there are SHL +2 and SHL -2 bound particles – the final maps could be at lower resolutions, nevertheless, it would be more precise. Even if this analysis shows no mixture of particles, we feel the analysis is necessary for conclusive interpretation of the data.Carrying out such an analysis would comprehensively address the following two prior questions asked by the reviewer.1) Upon comparing the PDB 3LZ0 with 6RYR, authors modeled 4 base-pairs on the exit site, which leads to the following question: The map EMD-10058_6RYR: If there is 4bp on the entry, and 30bp at the exit, upon filtering the density, why would there be such a large additional amount of density on the entry-side at the low threshold level? Could that be an indication of a mixed population in the final reconstruction, or a translocation?2) The authors write: "Structural comparisons show that CHD4, in contrast to Chd1, does not induce unwrapping of terminal DNA." The follow up to this is: "In contrast to the nucleosome-Chd1 structure (Farnung et al., 2017), we did not observe unwrapping of nucleosomal DNA from the histone octamer on the second DNA gyre at SHL -6 and -7 (Figure 2)". However, upon closer examination, this DNA density, while not unwrapped, appears to exhibit a lower occupancy than the rest of the nucleosome. Is that an effect of incomplete classification, or a partial distortion by the remodeler?

Note that our structure is at high enough resolution to support the unique assignment of the DNA register as shown in Figure 1—figure supplement 3H (purine and pyrimidine bases are distinguished, thus the DNA is uniquely assigned).

We have nevertheless followed the additional suggestion of the reviewer and have performed an analysis to test for possible symmetry artifacts. In short, CHD4 signal was subtracted from the Coulomb potential map and a masked refinement on the remaining NCP was performed with a synthetic NCP map as the model map. The particles were reverted to their non-subtracted state while maintaining the translational and angular information from the NCP refinement. Subsequently, a classification without image alignment was performed that resulted in clear density for CHD4 on only one side of the NCP. This clearly argues against a mixture of states where CHD4 is bound at SHL -2 and SHL +2 in the final reconstruction.

We added a corresponding paragraph to the Materials and methods section and have added these results to Figure 1—figure supplement 4F. Together with our high-resolution map, this entirely removes the concern.

[Editors’ note: further revisions were suggested prior to acceptance, as described below.]

[…]1) Addition of the following sentence in the text: "We cannot rule out that our map is still to some extent a mix of the CHD4 bound on either side of the nucleosome, as filtering of the map suggests the presence of more than 4 bp".

We have added the sentence "We cannot rule out, however, that our map is still to some extent a mix of CHD4 bound on either side of the nucleosome.” to the text.

2) Uploading the raw EM data to EMPIAR.

The raw EMPIAR data has been uploaded to the EMPIAR database with accession code EMPIAR-10411. This was already previously indicated in the last revision of the manuscript. The accession codes for PDB, EMDB, and EMPIAR datasets can be found in the data availability statement. All data sets will be made available upon publication.